# Opportunities and Challenges to Improve a Public Research Program in Plant Breeding and Enhance Underutilized Plant Genetic Resources in the Tropics

**DOI:** 10.3390/genes12101584

**Published:** 2021-10-09

**Authors:** Ivania Cerón-Souza, Carlos H. Galeano, Katherine Tehelen, Hugo R. Jiménez, Carolina González

**Affiliations:** 1Corporación Colombiana de Investigación Agropecuaria—Agrosavia, C.I. Tibaitatá, Mosquera 250047, Colombia; hjimenez@agrosavia.co (H.R.J.); cgonzaleza@agrosavia.co (C.G.); 2Corporación Colombiana de Investigación Agropecuaria—Agrosavia, C.I. Palmira, Palmira 763533, Colombia; galeanomendoza@gmail.com; 3Saint James Catholic Church, 1275 B St, Davis, CA 95616, USA; katherine.tehelen@gmail.com

**Keywords:** agrobiodiversity, food security, gender gap, germplasm banks, social networks

## Abstract

The American tropics are hotspots of wild and domesticated plant biodiversity, which is still underutilized by breeding programs despite being conserved at regional gene banks. The improvement of those programs depends on long-term public funds and the maintenance of specialized staff. Unfortunately, financial ups and downs complicate staff connectivity and their research impact. Between 2000 and 2010, Agrosavia (Corporación Colombiana de Investigación Agropecuaria) dramatically decreased its public financial support. In 2017, we surveyed all 52 researchers from Agrosavia involved in plant breeding and plant genetic resource programs to examine the effect of decimating funds in the last ten years. We hypothesized that the staff dedicated to plant breeding still suffer a strong fragmentation and low connectivity. As we expected, the social network among researchers is weak. The top ten central leaders are predominantly males with an M.Sc. degree but have significant experience in the area. The staff has experience in 31 tropical crops, and 17 are on the list of underutilized species. Moreover, although 26 of these crops are in the national germplasm bank, this has not been the primary source for their breeding programs. We proposed five principles to improve connectivity among teams and research impact: (1) The promotion of internal discussion about gender gaps and generation shifts to design indicators to monitor and decrease this disparity over time. (2) The construction of long-term initiatives and synergies with the Colombian government to support the local production of food security crops independent of market trends. (3) Better collaboration between the National Plant Germplasm Bank and plant breeding researchers. (4) A concerted priority list of species (especially those neglected or underutilized) and external institutions to better focus the collaborative efforts in research using public funds. (5) Better spaces for the design of projects among researchers and training programs in new technologies. These principles could also apply in other tropical countries with public plant breeding research programs facing similar challenges.

## 1. Introduction

Most of the biodiversity hotspots in the world are in the tropics. Moreover, this ecozone overlaps with 28% of the centers of crop domestication. Of these centers of domestication, tropical Latin America is critical. Some of the crops that originated and diversified in this region, such as maize, potatoes, and cassava, are of global importance. Many others are important only locally, however, thus maintaining a category of neglected and underutilized species (NUS) [1].

Despite these extraordinary conditions, Latin Americans face many intermixed problems such as poverty, food insecurity, and climate change [2]. Thus, Latin America’s future is challenging in a context where the human population will reach 9 billion people by 2050 [3]. Under these alarming conditions, world food production must increase from 25% to 70% compared with 2014 levels [3]. In this context, plant breeding programs in Latin America are critical, especially for NUS [4,5,6]. These species are critical to guarantee crop productivity (i.e., yield, resistance to diseases and pests), food and nutrient security, and specific crop adaptations to climate change (i.e., tolerance to heat, frost, drought, or salinity) [7,8,9,10]. Moreover, they could help to reduce infant mortality among poor households [11].

In Colombia, like many Latin American tropical countries, investment in plant breeding and plant genetic resource (PB&PGR) programs using regional agrobiodiversity has had ups and downs due to an economy at the mercy of internal conflict [12,13]. The peace agreement signed with the FARC guerrilla in 2016 recognized those historical problems. Therefore, it included public investment in better quality of living in rural areas, where agriculture is the primary income source, while strongly emphasizing the sustainable use of Colombia’s biodiversity [14,15]. Moreover, as a consequence of agrarian movements between 2013–2014, the national discussion about agriculture moved toward food provisioning in Colombia in a sovereignty context [16]. Therefore, public research programs associated with plant breeding require a high multidisciplinary and multistakeholder approach that the country lost after many years of lack of investment and should now reconstruct [17].

Colombia’s research in plant breeding began in 1955, directed by the DIA (Departamento de Investigaciones Agrarias). Many DIA experts developed the first successful plant varieties in different crops such as maize and potato (i.e., Diacol H104 and Diacol Capiro, respectively). In 1962, DIA became ICA, which started an essential collaboration with the Rockefeller Foundation, enhancing plant breeding, plant pathology, and plant physiology [18]. The ICA also generated grants and fellowships to support M.Sc. and Ph.D. studies and research training outside the country for many researchers. In addition to this, ICA led the creation and consolidation of the National Germplasm Bank of Colombia for agricultural purposes and several genetic improvement programs with a multidisciplinary research scope [19]. Both government initiatives contributed to establishing the first national master’s degree program in genetics and the first breeding program in collaboration with the Universidad Nacional de Colombia [20].

As a consequence of all this collaboration and investment in human resources, ICA produced and released many plant varieties during the next 30 years between the 1960s and 1980s, most notably maize, wheat, barley, oat, rice, and potato, which generated large impacts and benefits for producers in the country [20,21]. In the early 1990s, the government created Agrosavia (previously known as Corpoica) as a national research institution in agriculture, leaving ICA only in charge of the phytosanitary control, inspection, and surveillance functions. Unfortunately, between 2000 and 2010, government priorities changed toward a war-investment economy, and the research budget for agricultural research, including plant breeding programs, was severely affected [17]. Due to these decisions, the national plant breeding research goal shifted to carrying out field assessment of plant varieties developed by international centers such as CIAT (International Center for Tropical Agriculture), CIMMYT (International Maize and Wheat Improvement Center), and CIP (International Potato Center).

In 2014, the Colombian Congress approved Law Project No. 1731, which promised to assign an annual budget to support and promote the agricultural research programs in Agrosavia. Since then, Agrosavia has progressively renewed its research capacity by incorporating new research staff and new infrastructure. Currently, the central core of researchers working in plant breeding and plant genetic areas are young Ph.D. researchers trained in conventional and cutting-edge methodologies and tools to select and develop new plant varieties. However, Agrosavia is still experimenting with different strategies to accomplish its objectives. From 2014 to 2016, the main focus was creating research networks based on national agricultural production chains. Starting in 2017, Agrosavia has been working to consolidate disciplinary groups [17]. The goal of both plans was to renew the leadership in agricultural research using public resources. Therefore, in this study, we aim to identify significant gaps in the interactions between staff members and research challenges to create a policy to reactivate the plant genetic resources and public plant breeding research in Colombia lead by Agrosavia. Between 2000 and 2010, Agrosavia decreased its financial support, creating an internal competition for resources and using both Ph.D. and master degree staff mainly for external consulting [17]. As a consequence of this, we hypothesized that the staff dedicated to plant breeding and plant genetic resources (PB&PGR) still suffer a strong fragmentation and low connectivity, which leads to low impact in this area that researchers are able to perceive. Our analysis supported this hypothesis, and we therefore propose five principles that will help to reactivate public research leadership in PB&PGR led by Agrosavia.

## 2. Materials and Methods

In February 2017, we surveyed all 52 researchers from Agrosavia who worked on PB&PGR. We designed an online Google questionnaire divided into three sections: the group description, the challenges and opportunities in the PB&PGR area, and the social interactions within the PB&PGR group and with external researchers (Appendix A). We visualized the answers using the R-packages Plotrix 3.7–4 [22] and ggplot2 3.3.1 [23] in R Studio 1.2.5001.

### 2.1. The Group Description

We compared and visualized the number of researchers across age, sex, highest academic degree, years of experience, and rank category among the 52 researchers from the PB&PGR group in Agrosavia.

### 2.2. The Challenges and Opportunities in the PB&PGR Area

#### 2.2.1. List of Crops with Research Interest and Breeding Methods Used

We first ask to the 52 surveyed researchers to list all crops they had worked on in the last ten years. We then asked the researchers which crops they wanted to continue working on. The lists were compared and subsequently separated into three categories: increasing interest, equal interest, or decreasing interest. The researchers were also asked to report the plant breeding methods used for each crop listed. We used the list of crops from the surveyed researchers and compared it with three different lists of available information: the list released by Agrosavia between 2012 and 2021 of new varieties in 24 crops (Table 1); the list of species conserved in the Colombian plant germplasm bank that includes 275 crops and wild relatives (Reyes-Herrera, P. Pers. Comm, https://www.agrosavia.co/media/3542/lista_especies_bgv_2016.pdf, accessed on 1 September 2021); and the list of underutilized and marginal crops published previously in different research works from Colombia, the Andean region, and worldwide [4,5,6].

#### 2.2.2. The Connection between PB&PGR with the National Germplasm Bank and External Institutions

We counted the number of times the 52 researchers used the National Germplasm Bank in their projects, as well as the number and names of all external institutions with which they considered key to future collaborative work.

#### 2.2.3. The Identification of Strengths and Weaknesses in the PB&PGR Program

We asked the 52 researchers to indicate whether they strongly disagreed, disagreed, were neutral about, agreed, or strongly disagreed with the following ten statements associated with identifying the strengths and weaknesses across different activities and skills in the area:The PB&PGR group should participate in advising and training students and young professionals.The PB&PGR group should open spaces for journal clubs and research discussion.Students and young professionals have a good knowledge of quantitative genetics.Students and young professionals have a good understanding of fieldwork.PB&PGR researchers have a good understanding of MAS (marker-assisted selection).PB&PGR researchers have a good knowledge of experimental design.PB&PGR researchers have a good understanding of G (genetic) × E (environment) models.PB&PGR researchers have a good understanding of quantitative genetics.PB&PGR researchers have a good knowledge of molecular plant breeding.PB&PGR researchers have a good understanding of participatory plant breeding.

Finally, we had an open question asking for comments or suggestions to improve the PB&PGR group in the future. Because we obtained 31 different answers, we performed a corpus text mining analysis that included a sentiment analysis and a word cloud. The first step was to translate the 31 texts from Spanish to English using the free software DeepL (https://www.deepl.com/translator, accessed on 1 July 2019) and review them by hand. Then, we cleaned the corpus using the R-function gsub and the R-package tm 0.7–7 to remove punctuations, numbers, empty spaces, and stop words and to create a list of words associated with the workplace and research area (i.e., Agrosavia, plant breeding, plant genetic resources, etc.). Finally, using the R-package stringr 1.4.0, we broke each text into its characters to compare them with a positive and a negative list of words [24,25]. Both the positive and negative words were given a weight of one each. Thus, we generated a sentiment score for each text, subtracting the number of negative words from positive words. If the score was positive, the opinion had a positive sentiment, and vice versa. We calculated the average and the standard error of the sentiment scores across the 31 opinions. We also generated a word cloud based on the 30 most frequent words across the corpus using the R-package wordcloud 2.6.

### 2.3. The Construction of a Social Network Based on the Links within the PB&PGR Group and Externals

For this part of the survey, two researchers did not report any internal or external interactions. Therefore, we constructed a two-way social network for analysis consisting of 50/52 researchers from the PB&PGR group that specified 812 interactions within the group and 117 interactions with external researchers (i.e., either with other disciplines within Agrosavia and outside institutions).

We asked for six classes of interactions among researchers, such as collaboration on projects in the PB&PGR area, discussion about new advances in the PB&PGR area, germplasm interchange or request, preparation of scientific papers and grey literature, registration of new varieties, and technical advising. We compared the number of interactions between researchers from the PB&PGR group vs. between PB&PGR researchers and external researchers for each class. Additionally, within the network, we calculated and ranked the degree and the influence for each researcher (i.e., vertex). The degree measures the number of adjacent interactions (i.e., edges) under the R-package igraph 1.0.1 [26].

We compared the top-ten degree scores with the highest academic degree of the PB&PGR researchers to test the hypothesis that researchers with higher educational degrees will have higher numbers of interactions. Moreover, we measured each researcher’s influence based on two related statistics: the hub and the authority scores. The hub score measures the ability of a researcher to make a relation with other influential researchers (i.e., authorities). Thus, hub researchers are those that work together with trustworthy researchers on a common topic. In comparison, the authority score measures the number of links of interactions that a researcher earned. Therefore, authority researchers should have relevant information on the field and receive more links from other researchers. These two scores reinforce one another because a good hub hints at many competent authorities, and a proper authority is pointed to by many good hubs [27]. We associated the top ten of both hub and authority scores with the gender information of PB&PGR researchers to test the hypothesis that researcher influence is independent of gender.

We also calculated four descriptive statistics for the network (i.e., the width, the edge density, the average distance, and the transitivity) to diagnose how researchers work together within and outside the PB&PGR group. The diameter, the density, and the mean distance together indicate how connected the network is. In other words, they measure the ability of information to run through the system. Thus, for networks with fewer intermediates across interactions and higher direct connections, we expect lower diameter, higher density, and smaller mean distance. Likewise, the transitivity measures the local-scale structure of the network. Therefore, weak transitivity suggests that interactions occur in loosely connected clusters. In comparison, high transitivity indicates a well-consolidated system without a chance to identify discrete internal subgroups [28].

## 3. Results

### 3.1. The Group Description

We found a gender disparity among researchers working on PB&PGR within Agrosavia, where 71% (*n =* 37) are male, and 29% (*n =* 15) are female, and none declared an identity outside the binary male-female. Forty percent (*n =* 21) of male and female researchers are between 31 and 40 years old. Male researchers are present in all age ranges, but female researchers are more frequent at younger age ranges (Appendix A). Only two female researchers are in the 51–70 range, and none are older than 60 (Figure 1A). The survey showed that the ratio of researchers with M.Sc. and those with Ph.D. degrees is 2:1. Nine of the M.Sc. researchers are in the Associate or Senior category because they reached an outstanding productivity. Moreover, male researchers are present in all Agrosavia’s ranks and ages of experience within the PB&PGR group. In comparison, female researchers are only distributed in five of the eight categories, and have mostly 1–10 years of experience. No female researcher has more than 30 years of experience (Figure 1B). Concerning educational background, 52% (*n =* 27) are M.Sc. researchers (20 male and 7 female), and 27% (*n =* 14) are Ph.D. researchers (7 male and 7 female) (Figure 1B). When considering only lower academic degrees, both men and women studied only in Colombia. There is a gender imbalance associated with studying abroad, however. Of the 37 male researchers, 14 (9 M.Sc. and 5 Ph.D.) studied abroad. In contrast, of the 15 female researchers, only 4 obtained their Ph.D. degrees abroad (Figure 1C).

### 3.2. The Challenges and Opportunities in the PB&PGR Area

#### 3.2.1. List of Crops with Research Interest and Breeding Methods Used

Researchers from Agrosavia have a broad spectrum of investigation experience in at least 31 crops, of which 29 originated in the tropics, the only exceptions being soybean and blackberry (Appendix A). Moreover, of the 31 crops, 17 (55%) are considered NUS (Table 2). As a result, 40% (28 out of 69) of the new varieties released by Agrosavia since 2012 are in 8 of the 17 NUS (47%) (Table 2). They include beans, cocoa, maize, guava, pepper, potato, soursop, and sweet potato (Table 1).

Moreover, from this list of 31 species, 19 (61%) showed an increased research interest (i.e., more researchers interested in future projects than currently working on them), 3 species showed equal importance between past and future research plans, and 9 species showed decreased interest (i.e., few researchers interested in future projects than currently working on them) (Figure 2). Finally, the researchers have used clonal selection, mass selection, and marker-assisted selection more frequently than other breeding methods. New breeding methodologies such as genomic selection and genetic transformation are at the bottom of the list, with only one researcher with experience in each category (Figure 3, Appendix A).

#### 3.2.2. The Connection between PB&PGR with the National Germplasm Bank and External Institutions

Although the National Germplasm bank conserves 26 of the 31 species (84%) from the list in Table 2, the survey showed that this germplasm bank, also administered by Agrosavia, is not the primary focus of pre-breeding and breeding programs. A total of 37% (*n =* 19) of the researchers have never used the germplasm bank, and 40% (*n =* 21) have used it only once or twice in the last ten years (Figure 4). As a result, only 17% of the new varieties released since 2012 has at least one parent from an accession conserved in the National Germplasm bank (i.e., 8 new varieties of 69 registered) (Table 1, Appendix A).

Additionally, researchers working on PB&PGR are interested in maintaining or establishing collaborations with 54 universities and research institutions inside and outside Colombia (Appendix A). The most frequent was the Universidad Nacional de Colombia, with 23 researchers interested in collaborating. Moreover, 29 researchers want to have partnerships with three of the CGIAR (Consultative Group for International Agricultural Research) bases within and outside of Colombia, such as CIAT (Colombia), CIMMYT (Mexico), and CIP (Peru). CIAT and CIMMYT have previously partnered with Agrosavia to release new varieties of maize, bean, and grass (Table 1). Finally, Embrapa (Brazil), CIRAD (France), and Cenicaña (Colombia) had at least six researchers looking for collaborations (Figure 5).

#### 3.2.3. The Identification of Strengths and Weaknesses in the PB&PGR Program

We found that the researchers perceived weaknesses in fieldwork, marker assisted selection (MAS), experimental design, and participatory plant breeding skills (Appendix A). Moreover, the researchers felt that they should actively advise and train students and young professionals and lead in journal clubs and research discussion spaces (Figure 6). The opinion of 31 of the 52 researchers agreed on three main strategies to improve the PB&PGR program: (1) work as a network where the different PB&PGR researchers benefit from the knowledge of other members of the group in several areas and disciplines, (2) work jointly with other researchers that are strong in other subjects such as plant physiology and phytopathology, and (3) improve overall impact by focusing on very few strategic crops while developing a multidisciplinary long-term research program. The top five most used words included “programs”, “network”, “knowledge”, “improvement”, and “researchers” (Figure 7). The opinions and the most frequent words enclosed a positive sentiment score with a mean of 1.55 (0.3 SE) across 31 texts analyzed.

### 3.3. The Construction of a Social Network Based on the Links within the PB&PGR Group and Externals

The constructed network included 167 researchers: 50 from the PB&PGR group (with metadata associated) and 117 external, either from Agrosavia or other institutions (Appendix A). The 167 researchers had 812 two-way interactions among them. We found differences in the number of interactions depending on the types analyzed. The registration of new varieties had the lowest number of interactions (*n =* 50), whereas collaboration in projects in the area had the highest number of interactions (*n =* 242) within the network. Moreover, only two types of links among researchers (i.e., germplasm interchange or request and discussion about new advances in the area) showed more interactions within the PB&PGR group than between PB&PGR and external researchers (Figure 8).

Across the network, the diameter was 8, the edge density was 0.03, and the mean distance was 3.07. These values suggest a loose network with many intermediate researchers across interactions among researchers. Likewise, the transitivity was 0.17. This low value indicates a weak system characterized by several discrete and identifiable subgroups.

In the network center are the researchers with more influence (i.e., more interactions with other researchers). In contrast, in the periphery are mostly external researchers (Figure 9A). The top ten researchers with the most interactions (i.e., more degrees in the network) are all from the PG&PGR group, primarily males (*n =* 8) with an M.Sc. degree (*n =* 6) with 20 years of experience in the field (*n =* 7), who are planning to retire during the next two years (Figure 9B).

The top ten influencer researchers had a broader range of experience in the area and included three external researchers in the authorities list. However, similar to the degree score, they are primarily men (i.e., *n =* 8 in both groups), typically with an M.Sc degree (i.e., eight and six for hubs and authorities, respectively). Despite females being a minority as influencers, one female researcher obtained the top-ten score in both hubs and authorities (Figure 9C,D).

## 4. Discussion

### 4.1. The Background and Collaboration Networks

The growing world population, climate challenges, and loss of agrobiodiversity are the main focuses of plant breeders and plant genetic resource managers [1]. In this context, tropical crops, especially those categorized as NUS, offer an opportunity that requires researchers to work together using a transdisciplinary approach [4,5,6]. As we found in our survey, the Agrosavia researchers that work in PB&PGR have a high level of scholarly achievement and broad expertise in plant breeding, plant physiology, and plant production, among others. Although this experience is essential for helping to further develop and advance an effective research program over the following years, our analysis supported the hypothesis of a fragmented PB&PGR staff that perceived a low impact in its research. These gaps are associated with gender, generational shifts, and internal and external collaborations. Therefore, they represent areas that Agrosavia should focus on improving in the upcoming years to better impact our societies.

Our results showed strong gender bias in the composition of researchers working on plant breeding and genetic resources. Although the total female to male ratio in all of Agrosavia in 2017 was 0.613, it was only 0.408 in the PG&PGR group surveyed for this study [29]. Several studies focused on gender equality show that the agricultural sciences are a male-dominated field [30,31]. The disciplines of crop sciences, horticulture, and agricultural engineering worldwide have a female-to-male ratio of 0.435, 0.449, and 0.241, respectively [32]. Moreover, according to UNESCO, females working in agricultural and veterinary sciences make up only 37.6% of the research population [33]. This gender disparity is also evident in Colombia’s Agronomy and Veterinary sciences, where the female-to-male ratio is 0.769 for bachelor’s, 0.787 for M.Sc., and 0.694 for Ph.D. [34]. Moreover, the Agronomy Faculty from the Universidad Nacional de Colombia in Bogotá, which is the largest public university in the country and one of the main collaborators in PB&PGR, has a female to male ratio of 0.568 within students and 0.389 within professors [35]. For this reason, although faculty represents only 2.7% of the total population, it is the third most male-biased across all faculty groups in the university [36]. Thus, without any action, the workforce in Agricultural sciences in Colombia will not reach gender parity for approximately 25 years [37].

Also, our results showed that fewer female researchers in Agrosavia obtained their M.Sc. and Ph.D. degrees abroad than males. Although we did not have any questions in the survey that could directly determine the cause, other external indicators allowed us to hypothesize that females confront more social barriers in Colombia to studying overseas than males. Unpaid care and housework are strongly biased toward the female population in Colombia based on two indicators: the total number (89.5% of females vs. 62% of males assuming those responsibilities) and time (7 h per day spent on unpaid tasks for females vs. 3.5 h for males). Thus, 12.7% of females declared no free time compared with 8.1% of males [38]. Moreover, female Colombian scientists generally started their Ph.D. later than males, and typically already had unpaid care responsibilities [39]. Therefore, the combination of unpaid care and housework strongly biased toward females, a patriarchal academic background, and vertical and horizontal segregation while pursuing a scientific career may be hindering the mobility of women researchers to studying abroad [34,36]. Future studies about this topic in Agrosavia must include an intersectional gender and development approach to test this hypothesis [40,41]. This should include specific questions about sex and gender identity, the age for pursuing graduate studies, time spent in unpaid care and housework, and direct experiences of discrimination or barriers by gender, race, and class while applying for graduate studies abroad or at the beginning of their research careers, among others.

On the other hand, the network analysis allowed us to identify unbalanced distribution among the top-ten influencer researchers in PB&PGR by the highest degree and the sex [42]. Accordingly, the network showed a concentration of interactions (i.e., degree score) based mainly on male researchers with an M.Sc. degree and plenty of experience in the area. Additionally, these researchers showed the highest number of connections with other external actors, likely because their investigative journey has allowed them to expand the number of colleagues with whom they interact. Despite this, the research community is not yet well consolidated. As we hypothesized, direct collaboration is scarce. The productive subgroups are isolated and in the network’s periphery, generating loosely connected clusters with rare collaborations outside these subgroups (Figure 8). Although Agrosavia should focus on these key actors (nodes) as the best way to communicate and transfer innovation within the entire PB&PGR network, the main obstacle is that these identified leaders are close to their retirement age in the next four years. The network analysis also showed that Agrosavia researchers maintain strong links with many external researchers not included in the survey. This result suggests that those foreign actors are influential consultants for the design of a breeding program strategy. Therefore, the collaborative agreements that Agrosavia is constructing with several institutions inside and outside the country must encourage the formulation of projects to maintain and strengthen these known external collaborations.

The data collected in this study strongly support the urgency to have an internal discussion about improving diversity and inclusion during the upcoming generational shifts within the PB&PGR staff [34,37,39]. Currently, Agrosavia is a flourishing national institution for agronomic research. Therefore, promoting this discussion internally with a gender and development perspective would give Agrosavia specific indicators of gaps that could be monitored across time and corrected with explicit strategies (see this example for constructing indicators in conferences [43]). A policy supporting diversity and inclusion would also increase Agrosavia’s reputation, making it an attractive workplace, especially for researchers within the agronomic area that identify as women and other non-binary gender identities or sexual orientations. Moreover, it is an opportunity to ensure retention across the young Ph.D. researcher staff [37,44]. This internal discussion about gaps and inclusion could also provide an excellent opportunity to promote agricultural development with a gender perspective across different projects led by Agrosavia [45,46].

### 4.2. Research Experience and Skills in the Discipline

Researchers working on plant breeding and plant genetic resources in Agrosavia have two noticeable interests for current and future research: (1) underutilized tropical fruits for international markets and (2) food sovereignty and food security (Figure 2). However, as occurs in several countries, the current Colombian policies for improving food security and developing international markets do not necessarily identify NUS as a feasible alternative to achieve those goals [47]. Maize, for instance, is categorized as NUS in Colombia and is the top crop on the list in Figure 2. Many researchers are involved in maize breeding and genetics, seven new varieties have been released since 2012, and many more researchers are interested in working with it in the future [45,46,47,48,49,50,51]. This result agrees with previous reports because Colombia consumes 6.2 million tons of maize per year. Moreover, this crop is of critical importance for food security and the economy of small farmers [45,52,53]. Despite this, Colombia imports more than 80% of its maize, mainly from the United States and Argentina [54,55,56,57]. Moreover, no public maize breeding program currently focuses on open-pollinated varieties or hybrids to fill small farmers’ needs in Colombia and Latin America [58]. Although Mexico was facing the same problem as Colombia is today, where imports jeopardized the local maize production, they found a solution [48,49]. In 2010, the CIMMYT (International Maize and Wheat Improvement Center) and SAGARPA (Secretaría de Agricultura, Ganadería, Desarrollo Rural, Pesca y Alimentación) of the Mexican Government, successfully started MasAgro. This joint initiative aims to increase maize productivity, profitability, and sustainability [59]. The efforts of MasAgro significantly increased maize production and food self-sufficiency by implementing specific management practices, hybrid seeds, and direct sales in maize markets.

Designing a similar long-term goal in Colombia for maize and other NUS identified in this study requires synergies between the PB&PG public research and the Colombian government that guarantee effective marketing independent of global market trends [4,5,6]. Previous experience in Europe suggests that state funding is mandatory to generate competitive and economically viable NUS [50]. In this sense, PB&PG research in Agrosavia should focus on ranking the various NUS based on several indicators associated with market trends and previous research experience that we identified in this study. Then, PB&PG researchers should construct a collaborative roadmap using state funds for their research and incorporating new technologies like genomics and other phenomics approaches to determine the nutritional value and the adaptability of these NUS to several environments with a climate-change perspective [47,51,52]. Moreover, this research should include participative, communitarian, and gender-focused plant breeding approaches in concert with the government [47,53,54,55].

Furthermore, successful NUS breeding programs usually start with the use of a broad genetic germplasm base. Thus, collecting, conserving, and characterizing genetic resources are mandatory to introgress novel alleles into elite materials [52]. Colombia has a large National Plant Germplasm Bank with an extensive collection of underutilized native and introduced crops [60]. Three current factors are helping improve this situation in the future. First, since 2015, the National Plant Germplasm Bank has been working on an ambitious five-year project to implement a user-friendly a GrinGlobal platform for curators and users [61]. Second, as part of the same project, new approaches such as genomics are beginning to be used for a robust characterization of collections [53]. Currently, two critical crop collections, potato [62] and cacao [63], have been characterized and are available for the public. Soon, we expect to publish similar results for other native and underutilized crops such as avocado, Passiflora, and peach palm. Third, starting in 2018, Agrosavia was officially delegated by the Ministry of Agriculture to manage the Germplasm Banks for food and agriculture [17]. Therefore, we expect that a combination of active management, genomic characterization, high-throughput phenotyping, and genomic selection will increase the introgression accuracy of novel germplasm into a breeding program.

### 4.3. Perspectives for Future Alliances

This study also revealed the base of strategic alliances that Agrosavia is constructing with national and regional institutions focused on conventional crops (Figure 4). For instance, Colombia is validating technologies and varieties generated at other institutions, such as cassava, beans, forages, and rice from CIAT; maize from CIMMYT; and potato from CIP. Further, a new collaboration with high-ranking research institutions and universities such as Wageningen University, CIRAD, and the USDA opens new possibilities to co-lead international research initiatives using cutting-edge technologies and novel breeding approaches, which will facilitate the switch from being technology adopters to being knowledge and technology generators. Besides the current context of a knowledge-based economy where experience plays a vital role in economic growth, the growing relationship with universities as a critical player in the national innovation system becomes essential. An example of this is South Korea, which has evolved from being a developing state to a developed country [56]. Thus, the synergistic and collaborative work of Agrosavia with public and private universities has to be a priority. However, the innovation model also involves the industry in a triple helix paradigm where universities, industry, and government try to understand and cooperate between these components to boost national innovation performance [57].

The surveyed researchers expressed troubling issues such as generational inclusion, powerlessness, restricted capacities, and limited training opportunities in new areas (Figure 6 and Figure 7). Our results suggest that researchers are facing a change influenced by the new networking model and the entrance of at least seven new researchers in the area of genetics and plant breeding in the last four years. This situation can help explain individual resistance attitudes, which are usually governed by the anxiety caused by sudden changes in ages, experiences, scientific productivities, and abilities of the current plant breeding group at Agrosavia [58,59]. Although these results require more detailed analyses, the first approximation suggests that Agrosavia needs to quickly address this assortment of concerns to structure and consolidate a robust PB&PGR group, especially in genomic selection and genetic transformation areas.

This social network methodology could be extended in the future for the plant breeding group to manage resistance genes in different crops [60], to understand the dynamics of seed distributions [61], and to determine strategies in grain production [62]. Besides this, additional analyses using national survey and publication databases will provide a broad picture of how plant scientists in Colombia are collaborating in order to improve institutional governance and to increase and support these collaborative relationships [63].

## 5. Conclusions

This study is pivotal to understanding the challenges and opportunities for researchers in PB&PGR in Agrosavia, a state institution attempting to resume its leadership after suffering ten years of a lack of public financial support. Based on our analysis, we propose five strategies: (1) The promotion of internal discussion about gender gaps and generation shifts to design indicators to monitor and decrease this disparity over time. (2) The construction of long-term initiatives and synergies with the Colombian government to support the local production of food security crops independent of market trends. (3) Better collaboration between the National Plant Germplasm Bank and plant breeding researchers. (4) A concerted priority list of species (especially neglected and underutilized species) and external institutions to better focus the collaborative efforts in plant-breeding research using public funds. (5) Better spaces for creating and designing research projects and training programs in new technologies associated with plant breeding and plant genetic resource management. Furthermore, we suggest creating a consultative group to incorporate these five principles across high-impact research proposals to generate new cultivars that respond to national agriculture’s biotic and abiotic challenges. These principles could apply in tropical Latin American countries with public plant-breeding research programs facing similar challenges. Therefore, this study is also available in Spanish for further national and regional discussion in the same native language of researchers that generated the data (Appendix A).

## Figures and Tables

**Figure 1 genes-12-01584-f001:**
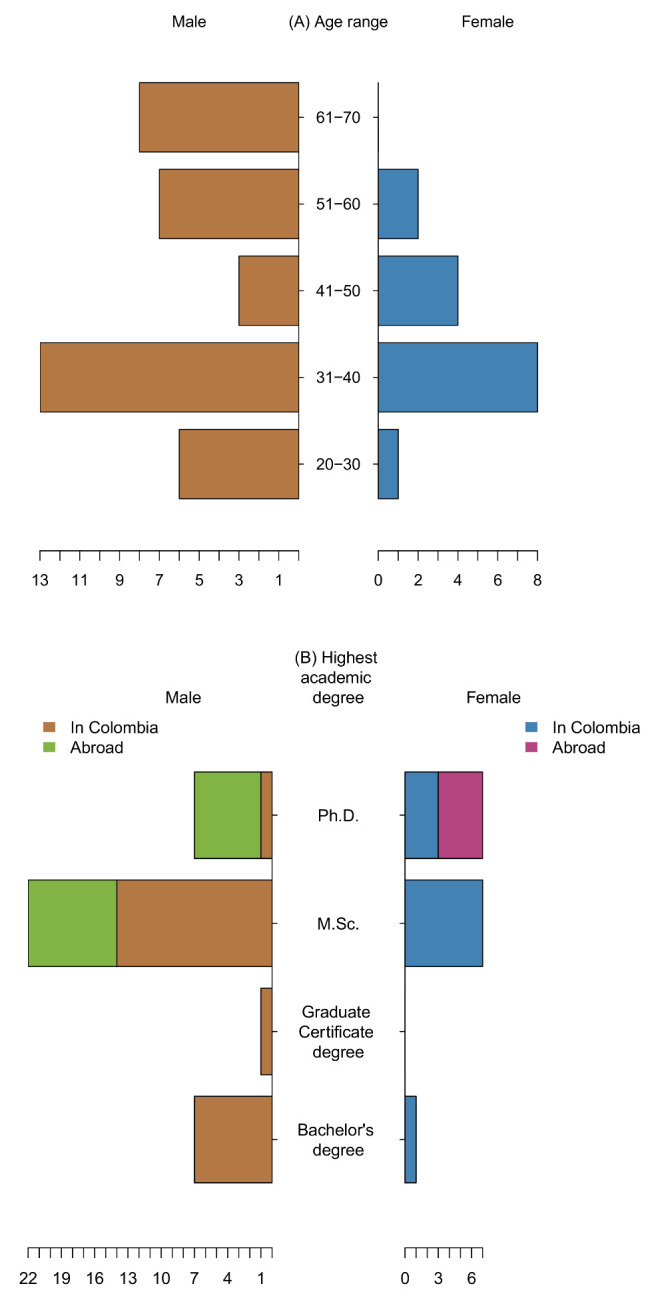
The characterization of 52 researchers working in Agrosavia that participated in this study. The number of researchers were categorized by (**A**) age range, (**B**) the highest academic degree obtained in Colombia and abroad, and (**C**) rank category in Agrosavia. The horizontal scales represent the number of researchers, with males to the left and females to the right.

**Figure 2 genes-12-01584-f002:**
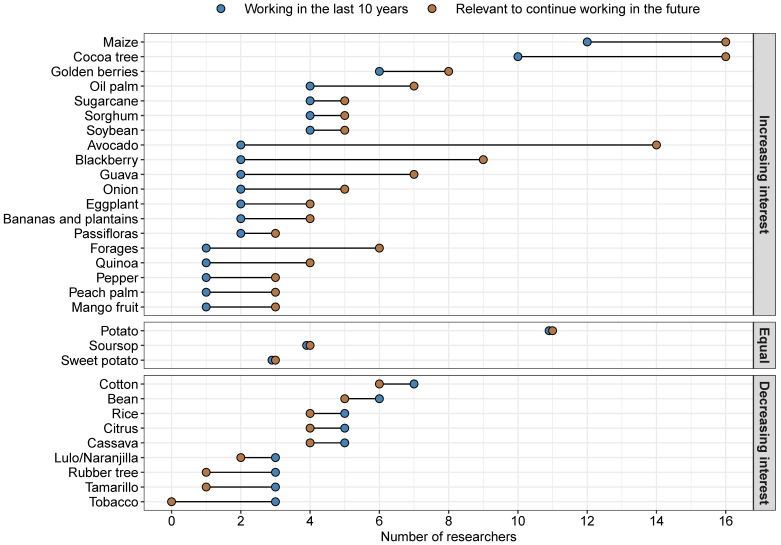
List of crops worked on by surveyed Agrosavia researchers in the last ten years (blue dots) and those considered relevant to continue working with in the future (yellow dots). Past and future research interests show three distinct patterns: crops with increasing interest for future work (increasing interest), crops with similar interest historically and in the future (equal), and crops with decreasing interest in the future (decreasing interest).

**Figure 3 genes-12-01584-f003:**
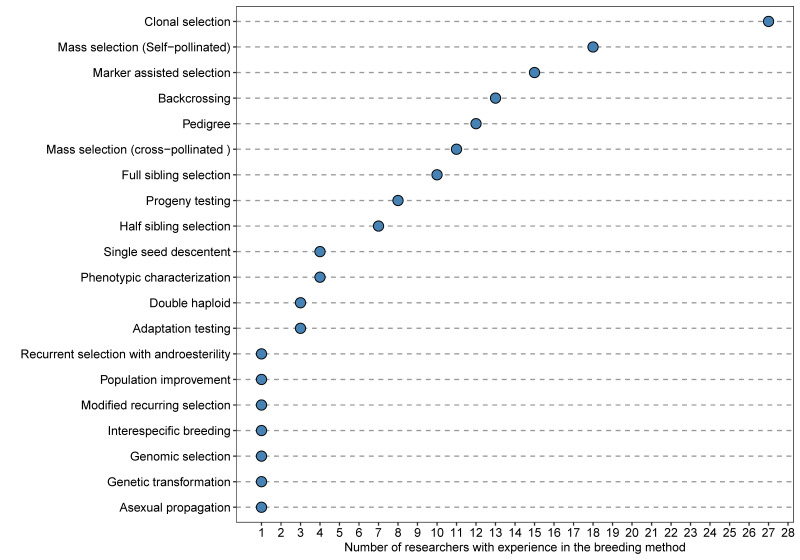
List of plant breeding methods used by researchers at Agrosavia in the last ten years sorted by the number of researchers (blue dots) with experience in each breeding method, from the highest to the lowest.

**Figure 4 genes-12-01584-f004:**
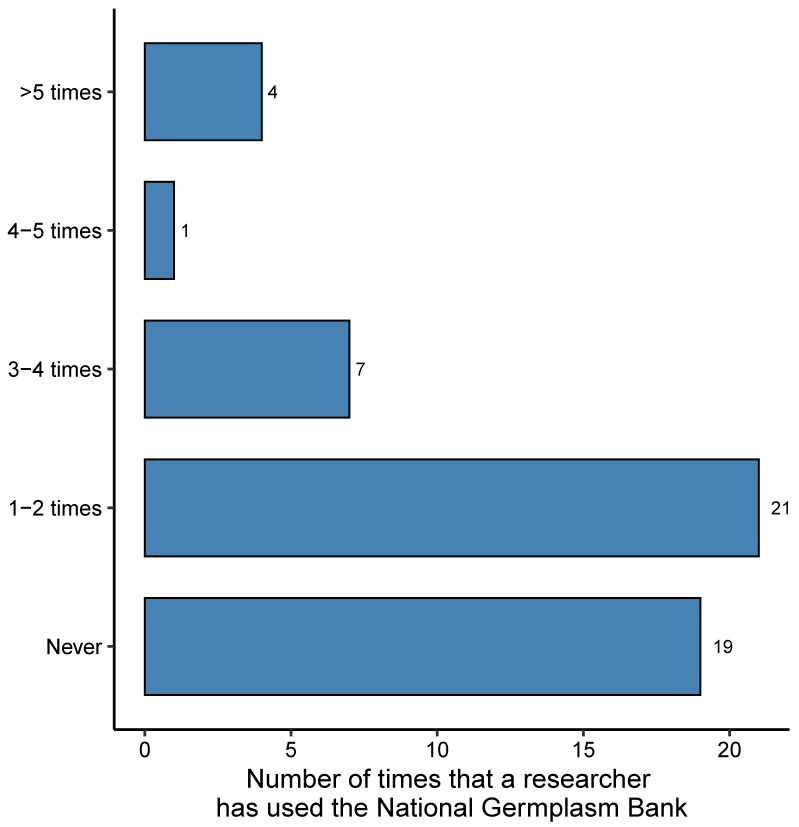
The number of times that researchers working on plant breeding and plant genetic resources from Agrosavia have requested germplasm from the National Plant Germplasm Bank of Colombia in the last ten years.

**Figure 5 genes-12-01584-f005:**
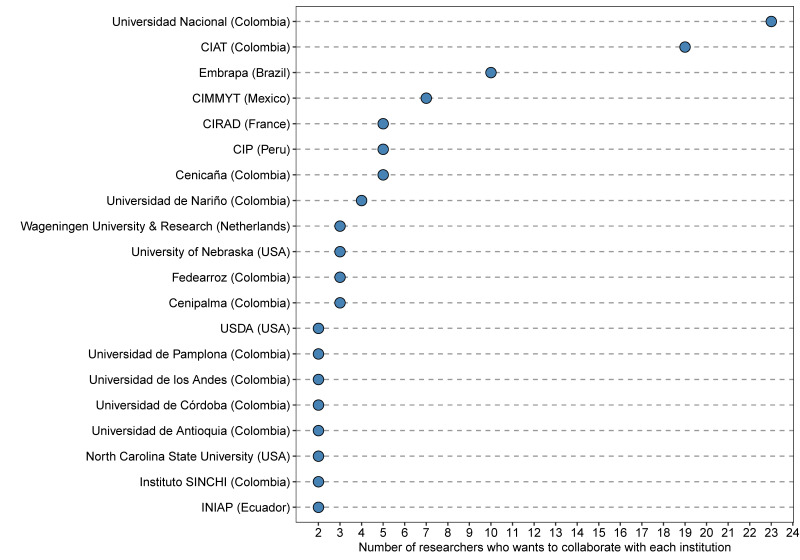
The number of researchers working on plant breeding and plant genetic resources from Agrosavia interested in starting collaborative breeding initiatives with each national and international research institution and university listed. In parenthesis is the country where each institution is located.

**Figure 6 genes-12-01584-f006:**
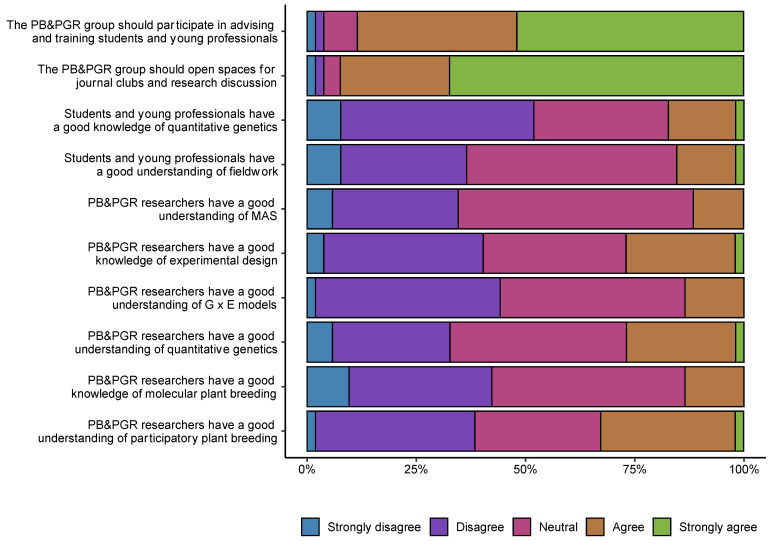
List of ten statements on challenges and opportunities associated with research in plant breeding and plant genetic resources within Agrosavia. The color bars represent the opinion of the 52 researchers surveyed, indicating in percentage the number of researchers who responded with each level of agreement for each statement. The categories from left to right are: strongly disagree, disagree, neutral, agree, strongly agree.

**Figure 7 genes-12-01584-f007:**
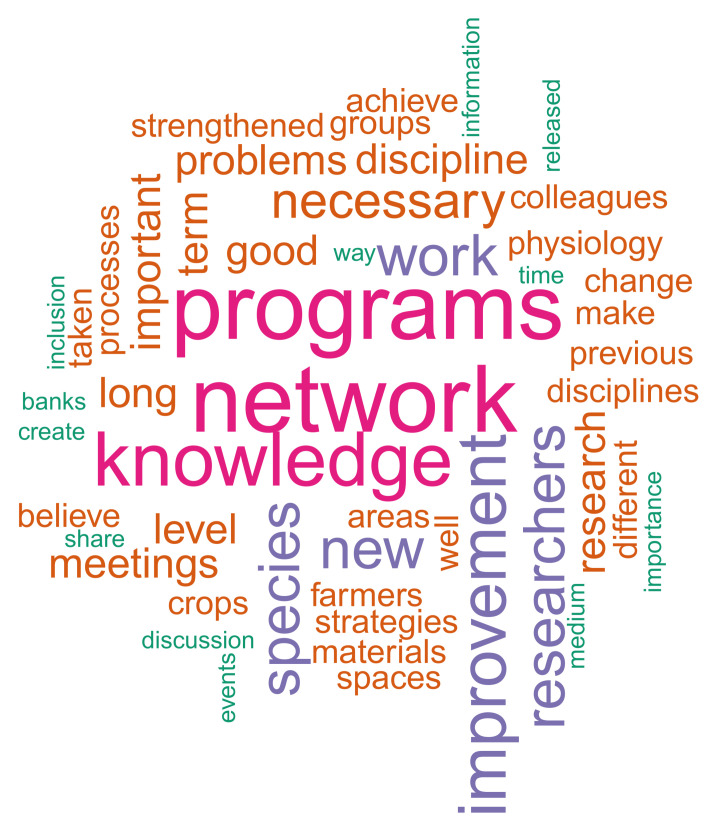
Graphic representation of a word cloud analysis of the 52 opinions about improving the plant breeding and plant genetic resources group in Agrosavia.

**Figure 8 genes-12-01584-f008:**
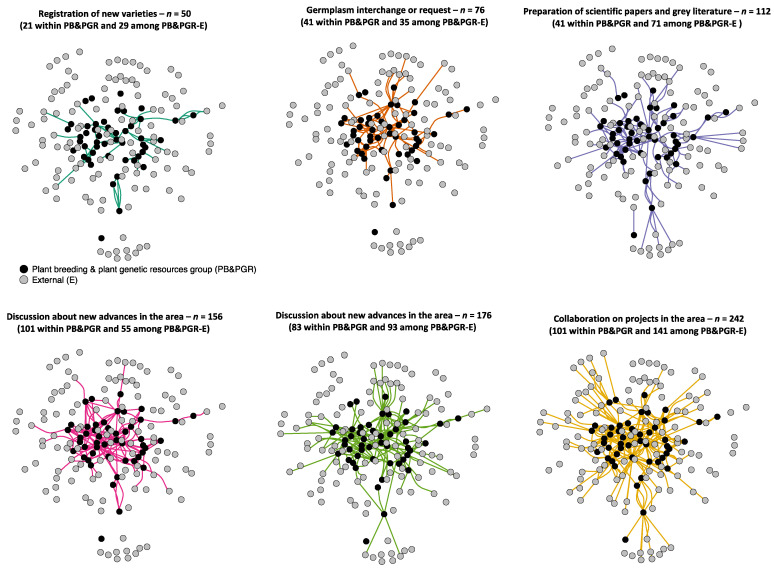
The six types of interactions among 50 researchers from the plant breeding and plant genetic resources group (PB&PGR) in black and 117 externals (including other disciplines within Agrosavia and outside institutions) (E) in grey. The six types of interactions are listed from the lowest number of total connections in the top left to the highest number of total connections in the bottom right. The number of interactions for researchers within the PB&PGR group and for researchers between the PB&PGR and externals is given in parentheses for each interaction type. The dots represent the researchers (i.e., vertex in the network), and the lines represent the interactions (i.e., edges in the system).

**Figure 9 genes-12-01584-f009:**
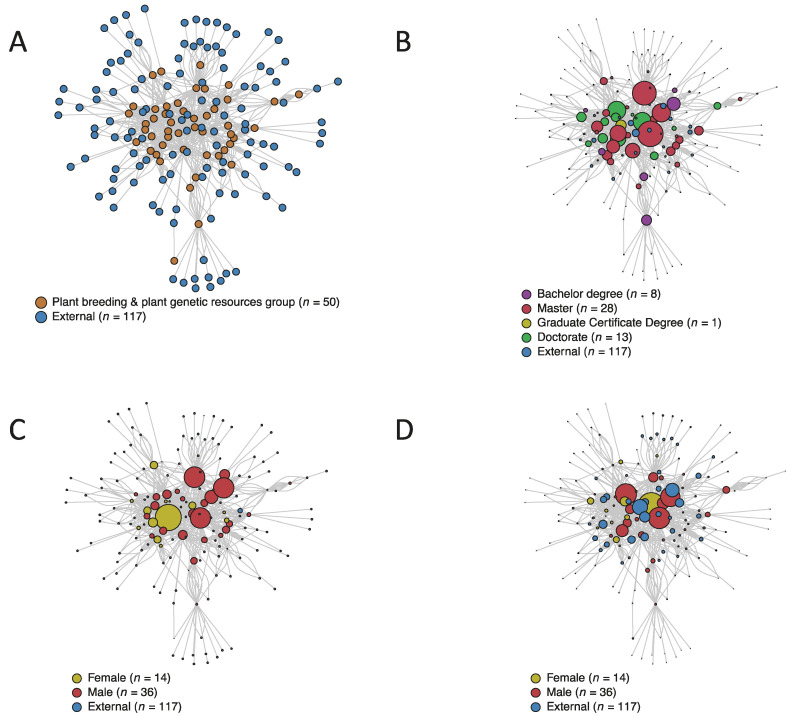
Graphical representation of the network analysis of 50 researchers working in plant breeding and plant genetic resources in Agrosavia and 117 externals (including researchers from other disciplines within Agrosavia and outside institutions). (**A**) Distribution of researchers (nodes) and their interactions (edges) separating the 50 researchers from the PB&PGR group and the 117 externals. (**B**) Characterization of researchers (nodes) by the degree score (i.e., size of the circle) and highest academic degree (i.e., color). The degree score indicates the number of adjacent direct interactions with other researchers. (**C**) Characterization of researchers (nodes) by the hub score (i.e., size of the circle) and gender (i.e., color). The hub score measures the ability of a researcher to make a relation with other influential researchers (i.e., authorities). (**D**) Characterization of researchers (nodes) by the authorities score (i.e., size of the circle) and gender (color). The authority score indicates the number of links of interactions that a researcher earned. Therefore, authority researchers should have relevant information on the field and receive more links from other researchers.

**Table 1 genes-12-01584-t001:** The number of crop varieties released by Agrosavia and registered in the national list of varieties (Registro Nacional de Cultivares, RNC—ICA) between 2012 and 2021 in 24 crops. It is noted that 4 varieties of the 69 were registered in collaboration with other institutions: 2 maize varieties with CIMMYT, 1 bean variety with Harvestplus and CIAT, and 1 grass variety with CIAT. Moreover, 7 of the 69 new varieties used as parent material at least one accession conserved in the Colombian Plant Germplasm Bank: 2 of onion and 5 of potato (Phureja group).

Crop Name	Scientific Name	Year of Release and Registration										
		2012	2013	2014	2015	2016	2017	2018	2019	2020	2021	Total Number of Varieties Developed per Crop
Arracacha	*Arracacia xanthorrhiza*	-	-	-	-	-	-	-	1	-	-	1
Bean	*Phaseolus vulgaris*	-	-	-	2	-	-	-	-	-	-	2
Cassava	*Manihot esculenta*	-	-	-	-	2	8	1	-	-	-	11
Cashew	*Anacardium occidentale*	-	3	-	-	-	-	-	-	-	-	3
Cocoa tree	*Theobroma cacao*	-	-	2	-	-	2	-	-	-	-	4
Cotton	*Gossypium hirsutum*	1	2	-	-	-	-	3	-	-	-	6
Golden berries	*Physalis peruviana*	-	-	-	-	2	-	-	-	-	-	2
Guava	*Psidium guajava*	-	-	-	-	-	2	-	-	-	-	2
Guinea grass	*Megathyrsus maximus*	-	-	-	-	1	-	-	-	1	-	2
Fodder Oat	*Avena sativa*	-	-	-	-	-	-	1	-	-	-	1
Maize	*Zea mays*	1	1	1	-	-	1	-	3	-	-	7
Oil palm	*Elaeis oleifera x Elaeis guineensis*	-	-	2	-	-	-	-	-	-	-	2
Onion	*Allium fistulosum*	-	-	-	-	2	-	-	-	-	-	2
Palisade grass	*Brachiaria brizantha*	-	-	-	-	-	-	-	-	1	-	1
Pepper	*Capsicum chinense*	-	-	-	-	-	-	-	-	-	1	1
Potato	*Solanum tuberosum*	-	-	-	-	1	-	-	-	-	-	1
Potato	*Solanum tuberosum* group Phureja	-	-	-	-	-	3	-	2	2	-	7
Rice	*Oryza sativa*	-	-	-	-	-	-	1	-	-	-	1
Sorghum	*Sorghum bicolor*	-	1	-	-	-	-	-	-	-	-	1
Soursop	*Annona muricata*	-	-	-	-	-	-	-	2	-	-	2
Soybean	*Glycine max*	-	3	-	-	-	-	-	2	-	-	5
Sugar cane	*Saccharum officinarum*	-	-	-	1	1	-	-	-	-	-	2
Sweet potato	*Ipomoea batatas*	-	-	-	-	-	-	2	-	-	-	2
Yellow peanut plant	*Arachis pintoi*	-	-	-	-	-	-	-	-	1	-	1
Total number of varieties developed		2	10	5	3	9	16	8	10	5	1	69

**Table 2 genes-12-01584-t002:** List of crops that researchers in Agrosavia work on or have worked with in the last ten years (see Figure 2), and whether or not they are listed as neglected and underutilized species (NUS) at three different levels: Colombia, Andes, and worldwide [4,5,6]. The crops are in alphabetic order showing the scientific name when applicable (NA—not applicable).

Crop Name	Scientific Name	Listed as Neglected and Underutilized Species (NUS)		
		Colombia	Andes	Worldwide
Avocado	*Persea americana*	YES	NO	NO
Bananas and plantains	*Musa* spp.	NO	NO	NO
Bean	*Phaseolus vulgaris*	NO	YES	NO
Blackberry	*Rubus* spp.	NO	NO	YES
Cassava	*Manihot esculenta*	NO	YES	YES
Citrus	*Citrus* spp.	NO	NO	YES
Cocoa tree	*Theobroma cacao*	NO	YES	NO
Cotton	*Gossypium* spp.	NO	NO	NO
Eggplant	*Solanum melongena*	NO	NO	YES
Forages	NA	NO	NO	NO
Golden berries	*Physalis peruviana*	NO	NO	NO
Guava	*Psidium guajava*	YES	YES	YES
Lulo/Naranjilla	*Solanum quitoense*	NO	NO	NO
Maize	*Zea mays*	YES	NO	NO
Mango fruit	*Mangifera indica*	NO	NO	NO
Oil palm	*Elaeis guineensis*	NO	NO	NO
Onion	*Allium cepa*	NO	NO	NO
Passifloras	*Passiflora* spp.	YES	YES	YES
Peach palm	*Bactris gasipaes*	YES	NO	YES
Pepper	*Capsicum baccatum*	NO	YES	YES
Potato	*Solanum tuberosum*	NO	YES	NO
Quinoa	*Chenopodium quinoa*	YES	YES	YES
Rice	*Oryza sativa*	NO	NO	NO
Rubber tree	*Hevea brasilensis*	NO	NO	NO
Sorghum	*Sorghum bicolor*	NO	NO	NO
Soursop	*Annona muricata*	NO	YES	YES
Soybean	*Glycine max*	NO	NO	NO
Sugarcane	*Saccharum officinarum*	NO	NO	NO
Sweet potato	*Ipomoea batatas*	NO	YES	NO
Tamarillo	*Solanum betaceum*	YES	NO	YES
Tobacco	*Nicotiana tabacum*	NO	NO	NO

## Data Availability

All data generated in this study are available in the Appendix A changing names and personal information for a researcher number.

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
