# Peer review of "Opportunities and Challenges to Improve a Public Research Program in Plant Breeding and Enhance Underutilized Plant Genetic Resources in the Tropics"

_genes, 2021, doi:10.3390/genes12101584_

Round 1

Reviewer 1 Report

[General]

This article analyses results of questionnaire of researchers in AGROSAVIA aiming to identify opportunities and challenges of research in PB&PGR of the institute. Based on the survey results, authors identified strengthen and weakness of the PB&PGR of the AGROSAVIA. The approach resembles to “Evidence-based policy making” and the application for institutional policy making is novel. The research would be of interest to other research programs.

In the Conclusion, authors proposed five principles to improve the research impact. From now on, research direction of the PB&PGR in AGROSAVIA will be decided based on these five principles. It is important to verify the research output which to be developed based on these principles in the future.

[Keywords]

Please reconsider Keywords. I feel that “diagnosis” is too general as a Keywords for a research article. Since the listed keywords are methodological words only, I recommend adding words linked with the objective/outcome of this research.

[Materials and Methods] 

I'm not requesting to add new data to this research, but in addition to the questionnaire, adding quantitative data of past/current PB&PGR activities (e.g. number of released varieties, number of domestic genetic resources used etc.) would make the discussion on the importance of domestic genetic resources more persuasive.

[Discussion]

1) Page 15 Line 359-375

Description of sex ratio of the researchers in this paragraph should be uniform (Line 361 male/female; Line 364 female/male; Line 371 male/female).

2) 4.2 Research experience and skills in the discipline

Page 16 Line 426-448

Discussion on the underutilized tropical fruits should be added here. Adding the economic scale, global market potential, and farmers' needs will make the importance of the underutilized tropical fruits for international markets more persuasive.

[Figures]

The color used in the Figure 1B and 1C should be reconsidered. Especially gradation of reddish color is not distinguishable. Letters of the Figure 1 is too small and not visible. Size of Figures should be rearranged.

Author Response

[Keywords] Please reconsider Keywords. I feel that “diagnosis” is too general as a Keywords for a research article. Since the listed keywords are methodological words only, I recommend adding words linked with the objective/outcome of this research.

Done. We eliminated the word “diagnosis”. The new keywords are agrobiodiversity; food security; gender gap; germplasm banks; social networks.

[Materials and Methods] I'm not requesting to add new data to this research, but in addition to the questionnaire, adding quantitative data of past/current PB&PGR activities (e.g. number of released varieties, number of domestic genetic resources used etc.) would make the discussion on the importance of domestic genetic resources more persuasive.

Thanks for this suggestion. In this section, we included Table 1 containing the number of released varieties between 2012 and 2021 by Agrosavia and the link to access the list of species conserved in the germplasm bank that includes 275 species of crops and wild relatives.

We used this information in two sections (3.2.1. and 3.3.2) to compare the number of release varieties with the answers from the survey. This comparison shows us that 47% of new varieties Agrosavia has released since 2012 correspond to eight species classified as NUS (neglected and underutilized species) and includes beans, cocoa, corn, guava, pepper, potato, soursop, and sweet potato. In contrast, the researchers are not using the accessions conserved in the national bank as a source of parents to develop new varieties. From the 69 new varieties released, only eight new varieties registered that at least one of the parents is an accession conserved in the National Germplasm bank.  Finally, this comparison shows that Agrosavia has released three new varieties in collaboration with CIMMYT and CIAT, demonstrating the importance of maintaining external collaborations.

 [Discussion]

1) Page 15 Line 359-375

Description of sex ratio of the researchers in this paragraph should be uniform (Line 361 male/female; Line 364 female/male; Line 371 male/female).

Done. We unified all the sex ratios showing female/male.

2) 4.2 Research experience and skills in the discipline

Page 16 Line 426-448

Discussion on the underutilized tropical fruits should be added here. Adding the economic scale, global market potential, and farmers' needs will make the importance of the underutilized tropical fruits for international markets more persuasive.

Thanks for this suggestion. We added new references in this section to mention what should be the steps to find new international markets based on the experience in other countries. Based on these results, we emphasized the need to state funding to generate competitive and economically viable NUS and the importance of involving the communities.

[Figures]

The color used in the Figure 1B and 1C should be reconsidered. Especially gradation of reddish color is not distinguishable. Letters of the Figure 1 is too small and not visible. Size of Figures should be rearranged.

We changed the colors of this figure to generate more contrast and put the parts A, B, and C, separately to display an appropriate letter size and original figure size.

Reviewer 2 Report

the manuscript is well written and original, although it doesn't sound very interersting for the people out of Colombia and South America. Anyway it provides several interesting data representing inspiration for some considerations which may be applied to different realities, firstly the "fragmentation" of the reserach staff perceiving a low impact in research. One of the most important aspect, which perhaps should be more thoroughly investigated is the importance of study tropical crops (NUS), which require a trans-disciplinary approach. The discussion and conclusion are coherent with the setting of the paper and the and with the answers to the interviews.

Author Response

The revised version was revised for a native English speaker.